# Research on the Preparation of Biochar from Waste and Its Application in Environmental Remediation

**Wanyue Wang** [1], **Jiacheng Huang** [1], **Tao Wu** [1], **Xin Ren** [1,2,*] **and Xuesong Zhao** [1,2,*]

1   Key Laboratory of Environmental Materials and Pollution Control, Education Department of Jilin Province, Siping 136000, China; 15834814573@163.com (W.W.); 15981436266@163.com (J.H.); tao20010604@163.com (T.W.)
2   College of Environmental Science and Engineering, Jilin Normal University, Haifeng Street, Tiexi Dist., Siping 136000, China
*   Correspondence: renxin_hit@126.com (X.R.); zhaoxuesong008@163.com (X.Z.); Tel.: +86-434-3291050 (X.R.)

**Highlights:**

What is the main research?

- Biochar production using multiple waste streams as feedstock.
- Introduction to different modification methods.

What is the significance of the main discovery?

- Biochar and its composites in environmental applications.
- The existing shortcomings of biochar materials and their outlook for the future.

**Abstract:** Biochar is a carbon-rich material that can be composed of a variety of raw materials. From the perspective of resource reuse, it is quite feasible to use waste as a raw material for the preparation of biochar. This paper provides an overview of the types of waste that can be used to prepare biochar and their specific substances, and also summarises methods to enhance or improve the performance of biochar, including physical, chemical, biological and other methods. The feedstock for biochar includes four categories: agricultural and forestry waste, industrial by-products, municipal solid waste and other non-traditional materials. This paper also summarises and classifies the role played by biochar in environmental applications, which can be classified according to its role as an adsorbent, catalyst and soil conditioner, and other applications. In addition to being widely used as an adsorbent, catalyst and activator, biomass charcoal also has good application prospects as a soil remediation agent, amendment agent and supercapacitor, and in soil carbon sequestration. Finally, some ideas and suggestions are detailed for the present research and experiments, offering new perspectives for future development.

**Keywords:** biochar; biochar modification technology; waste; resource reuse; environmental applications; catalyst and activator

## 1. Introduction

Biochar is a thermogenic organic material synthesised via the pyrolysis of different biomasses [1]. There are many organic wastes, such as agricultural waste and municipal solid waste, that are used as raw materials for the production of biochar. Biochar is attracting more and more attention because it has abundant carbon, a strong ion swap ability, a high specific surface area and a stable structure [2]. The production of biochar not only treats waste, but also makes it profitable [3]. The use of waste biomass to produce the corresponding biochar also realises the concept of resource reuse.

There are many types of waste biomass that can be used to produce biochar, and waste can be categorised as agricultural, forest, industrial or municipal solid waste, as well as that

from other sources. Firstly, waste from agriculture has the advantage of being productive and easily accessible. Take straw and husks of crops as an example: MS corn straw and RH rice husk can be used to prepare biochar [4]. Secondly, biochar prepared from forestry waste wood has good physicochemical properties (large surface area and aroma properties, among others) and therefore has great potential for some aspects of water treatment applications [5]. Fe-Mg layered double hydroxide (LDH) is widely dispersed on cheap, commercially available Douglas fir biochar, and this mixed multiphase LDH dispersed on biochar strongly adsorbs phosphate from aqueous solutions with exceptional adsorption capacity [6]. Once again, by-products from industry can be used as raw materials for the preparation of biochar, for instance, the preparation of biochar and adsorption of Cu(II) from sludge in municipal wastewater treatment plants [7]. Similarly, studies have been conducted using paper sludge/wheat hull biochar to adsorb 2,4-dichlorophenol [8]. This is followed by municipal solid waste, which consists mainly of municipal solid waste mixtures, food waste and recyclable waste (such as waste paper and tyres), all three of which can be used to produce biochar [9]. Lastly, there are the non-traditional ingredients that are distinguished from the first four. These include raw materials such as algae and animal manure. Experimental studies have been carried out to prepare biochar from algae as an efficient dye adsorbent [10].

In order to better apply biochar, modifications are used to improve its various physicochemical properties. The modifications or activations can be carried out in a number of ways, but these can be broadly divided into two areas. On the one hand, the object of modification is the pretreatment of the raw material and the modification of the original biochar [3]. On the other hand, the means of modification are mainly divided into physical methods [11], chemical methods [12], biological methods [13] and using biochar-based composites [14], among others. These treatments include steam, gas purification [15], microwave, magnetic, acids, bases, metal oxides [16], hydroxides and bacteria. The changes resulting from this treatment are broadly (1) improved physical properties, (2) improved chemical properties and (3) the use of biochar as a substrate for a wider range of materials [14]. The modified biochar can improve its physicochemical properties and enhance its ability to remove pollutants, while being more suitable for industrial applications for promotion. If analysed from another perspective, biochar modification can be classified into surface doping and surface composite. The former can not only change the surface properties, but also give special functions to the biochar material, while composite is the use of chemical reactions and functional groups to improve the performance of biochar [17].

Biochar has a large specific surface area, is difficult to degrade, is catalytic and has a wide range of applications in the environmental field, including soil remediation, water treatment and climate change, in the reduction in greenhouse gas emissions [18]. Biochar plays a specific role as an adsorbent, catalyst, soil conditioner and so forth. Regardless of its role, the main pollutants treated by using biochar can be classified as inorganic, organic and radioactive [19]. As a soil remediation agent, it not only removes inorganic and organic pollutants, but also improves acidic soils [20], enhances soil fertility for crop growth, fixes carbon and has many other applications. Biochar produced from wood and switchgrass can exhibit high carbon sequestration potential when the operating parameters for pyrolysis are properly selected [21]. Overall, the type of feedstock and preparation conditions are determining factors for the productivity of biochar [22]. The main application of biochar is for pollutant removal, followed by climate change mitigation, soil fertility improvement, waste management and atmospheric carbon sequestration into the soil [23]. Then, when biochar is used to remove pollutants, this mainly relies on the functional groups of biochar, such as carboxyl and hydroxyl groups, which have different degradation mechanisms for different pollutants, some due to ion exchange and others due to surface complexation [24]. Although biochar has a wide range of promising applications in environmental remediation, detailed experimental information is still needed to conduct pilot-scale studies of biochar in soil amendment and climate change mitigation. In summary, the possibility and utility

of biochar both as an adsorbent material and as a catalyst in future research is of high value, so the global importance of biochar and the necessity of this study is self-evident [25].

The aim of this paper is to summarise, using previous reviews on biochar, the progress of research on raw materials for the preparation of biochar, its modification methods and its various resourceful reuses as an adsorbent and catalyst in environmental terms. It also suggests aspects of biochar that deserve consideration in practical applications. Finally, based on the summary and comparison of previous studies, suggestions are made for future research.

## 2. Waste Biomass Feedstock

Biochar is a biomass-derived functional material obtained by the thermochemical conversion of biomass [26]. Of these four different sources of materials, agricultural and forestry wastes are the most readily available and abundant, so the advantages of biomass charcoal produced from them are ease of preparation, large quantities, and low production costs at low prices. Industrial by-products and other by-products generated in the production process can also be used to generate additional income for the factory, which is a greater realisation of the reuse of waste resources. The variety of municipal solid waste and other non-traditional materials is not conducive to uniform treatment. The raw materials for biochar include four categories: agricultural and forestry waste, industrial by-products, municipal solid waste and other non-traditional materials [27]. Three of these types—agricultural and forestry waste, industrial by-products, and municipal solid waste—are all characterised by high quantities and easy accessibility. However, if we compare them carefully, industrial by-products and agricultural and forestry wastes are easier to convert into biochar, while municipal solid wastes and other non-traditional materials are not easily processed uniformly and quickly due to their great variety. At the same time, agricultural and forestry waste and industrial by-products can also be reused as waste resources, providing a portion of the objective income for the producer or enterprise. Figure 1 shows the sources of different biochar materials.

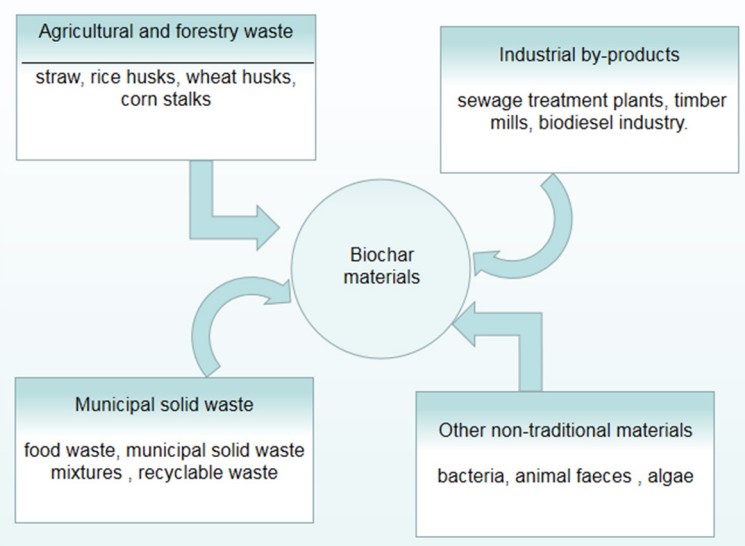

**Figure 1.** Sources of different biochar materials.

### 2.1. Agricultural and Forest Wastes

In 2017, the top three cereals produced in the world were maize, rice and wheat. Of these, world wheat production was 607 million tonnes. The rice industry is similar to the wheat industry in that it generates some waste, such as wheat hulls [28]. These three food crops are hugely productive, and there is also an abundance of agricultural waste from these three crops: straw, rice husks and wheat hulls. In 2020, for example, one study

concluded that the total amount of collectable straw in China for maize, rice and wheat was 21.597 million tonnes, 141.809 million tonnes and 13.573 million tonnes, respectively [29].

The agricultural waste from the three major food crops mentioned above is used as the main raw material for the preparation of the corresponding biochar. Firstly, maize was used to prepare biochar using maize stover biomass heated to 400 °C. The composite BC-LDH, a composite of biochar and Co-Fe layered double hydroxide, was also used to enhance the activation of persulfate, and the composite BC-LDH showed superior catalytic activity and stability in PMS activation [30]. Secondly, rice can provide two types of waste biomass, one of which is biochar from rice husks. Some studies have used this biochar prepared from rice husks as a substrate for loading perylene dimethyl ether (PDI/BC, PB) as a photocatalyst [31]. The second type is biochar produced from rice straw for the efficient degradation of estrone by leaching bismuth biochar [32]. Finally, there is wheat, which provides similar waste biomass to rice. On the one hand, wheat hulls are used to prepare biochar. There are experiments using wheat hulls and paper sludge together to prepare biochar, which has been combined with Zn-Co-LDH to obtain a composite material for the photodegradation of gemifloxacin antibiotics [33]. On the other hand, biochar has been prepared from wheat straw. $CeO_2$–biochar composites have been used for the ultrasonically catalysed degradation of textile dyes, in which biochar was prepared from wastepaper and wheat straw [34]. Wheat, the world's third-largest cereal, is also a great source of biochar feedstock due to its malt roots. It can be used to make supercapacitor electrodes, which have increased supercapacitance values because of the material's good specific surface area [35,36]. Biochar made from agricultural waste has the advantage that raw materials are easily accessible and abundant. At the same time, the carbon content of straw is around 40%, which makes it a good raw material for the production of biochar [37]. In addition to the main crops mentioned above, there are many other materials that can be used to prepare biomass. For example, peanut shells, bagasse, and its biochar can be used as adsorption materials and show good adsorption properties [38,39].

There are many raw materials available in the forest for the preparation of biochar, for example, reeds and invasive plants. Experiments have been conducted to prepare biochar from reed straw and combine it with $Bi_2WO_6/Fe_3O_4$ to obtain $Bi_2WO_6/Fe_3O_4/BC$, a composite material with high photocatalytic activity [40]. Using invasive plants to prepare biochar has also been proposed, which has many advantages: saving production costs, increasing invasive plants' value and extending it to a wider range of fields, and presenting an efficient way of using resources [41]. The production of biochar from these two raw materials not only reuses resources, but also protects the ecology of the forest.

### 2.2. Industrial By-Products

Industrial by-products come from sewage treatment plants, timber mills and the biodiesel industry. Sludge from wastewater treatment plants can be used as a raw material for biochar production, and some studies have used sludge-derived biochar to improve soil [42]. The feedstock available from wood mills can be divided into two forms. One is in powder form, such as in a 2019 article using poplar sawdust and dicyandiamide as feedstock to create nitrogen-doped biochar for the activation of peroxymethane sulphate (PMS) in wastewater treatment [43]. The second is in flake form, and there has been research into the photocatalytic degradation of antibiotics using biochar prepared from poplar woodchips collected in timber mills [44]. In addition to the two industries mentioned above, there are good feedstock options in the biodiesel industry. One is the preparation of biochar from algal residues, a waste product of the biodiesel industry, as an adsorbent for the removal of dyes from water [45]. The application of seaweed residues to produce biochar enables the reuse of waste resources and also supports the sustainability of algal biodiesel. To summarise the abovementioned article, biochar prepared from raw materials from industrial by-products can be used as a soil conditioner, catalyst and adsorbent.

### 2.3. Municipal Solid Waste

Municipal solid waste (MSW) generation is growing at an annual rate of 8–10%, with over 150,000,000 tonnes of MSW generated each year [46]. Municipal solid waste (MSW) is material discarded in urban areas and includes paper, food waste, leather, plastics, rubber and fabrics, to name a few. These are all sources of biomass [47]. There are many different types of municipal solid waste, and these are now divided into three main types: food waste, municipal solid waste mixtures and recyclable waste.

The annual production of municipal solid waste in the world is about 4 billion tonnes, of which food waste and rotting matter, which make up the largest share, account for roughly 30–45% of the total, depending on the region and country [48]. Based on experiments, food waste sampling can be divided into seven types: meat and bones, starchy staples, vegetables, nut shells, fruit peels, soya bean residues and tea leaves [49]. As higher-carbon-content feedstocks are suitable for biochar preparation, meat and bone have less carbon content than other feedstocks, and their high ash and mineral content combine to make them unsuitable as feedstocks for biochar preparation. Among other classifications, there are studies using pepper stalks as a feedstock for biochar. The biochar produced was also combined with $CuFeO_2$ to degrade tetracycline and improve catalytic performance [50]. Biochar derived from food waste can be used in environmental applications not only as a catalyst, but also as an adsorbent. When biochar is prepared from food waste and used as an adsorbent, the effectiveness of the adsorption varies depending on the raw material. Of these, raw materials such as fruit peels and nut shells have a good adsorption effect, and the least effective is biochar made from starchy staples, meat and bones [49].

Municipal solid waste mixtures are mainly household waste, waste plastics, organic matter, metals and other materials such as waste glass [51]. Mixed municipal solid waste can be pyrolysed into a higher-value product, biochar, enabling the resourceful use of the waste [9]. The properties of biochar are related to the raw material, and because of the variety in municipal solid waste mixtures, biochar can have different physicochemical properties. Some studies have confirmed that municipal solid waste biochar (MSW-BC) can remove volatile organic compounds (VOCs) from water [50]. Some of the characteristics of mixed household waste are similar to those of the meat and bone materials in food waste discussed above in that they both have a high ash content. The high ash content of meat bones makes them unsuitable for biochar preparation, and similarly, the high ash content of MSW is not the best choice for the preparation of high-specific-surface-area biochar. It is also due to the complexity of its feedstock that the microporous structure of mixed municipal solid waste-derived biochar is random and disordered [9]. Municipal solid waste mixtures have a bright future because of their ease of handling and availability, and the abundance of raw materials.

Recyclable waste includes other materials such as wastepaper, waste plastics, waste glass and others. One experiment used wastepaper and wheat straw to prepare biochar, combined with $CeO_2$, to degrade textile dyes [34]. The surface of plastic-derived biochar is smoother and has less surface area than that of paper-derived biochar, resulting in poorer adsorption performance and making it unsuitable for water treatment [52].

### 2.4. Non-Traditional Materials

In addition to the above types of materials, there are other materials that can be used to prepare biochar, and these are classified as non-conventional or non-traditional materials. Non-traditional materials include bacteria, animal faeces and algae. The first is a bacterial feedstock, and experiments have been conducted using Mn combined with bacteria-derived biochar to activate peroxymonosulphate [53]. Secondly, animal manure is used as a raw material, including a wide range of manure such as cattle manure, poultry manure [54], pig manure and earthworm manure, among others. Using earthworm dung as an example, one study used modified earthworm dung biochar as an adsorbent to adsorb cadmium [55]. The last category is the preparation of biochar from algae as a feedstock, and there are studies arguing that biochar from microalgal biomass collected from wastewater treatment

systems could be used as a soil amendment or sorbent in the future [56–58]. Non-traditional materials are not as widely used as materials in the previous categories, but the future looks bright. In summary, the raw materials of biomass charcoal are mainly divided into five categories, but even for materials from the same source, the preparation process is not the same: the actual application of the material needs to be based on the material's specific properties to control the vegetation time and temperature. Table 1 shows various feedstocks and methods for biochar preparation.

**Table 1.** Various feedstocks and methods for biochar preparation.

| Type of Biomass | Feedstock | Preparation Technology | References |
| --- | --- | --- | --- |
| Crop residue | Maize straw | Pyrolysis, 400 °C, 2 h | [30] |
| | Rice husk | Pyrolysis, 500 °C, 2 h | [31] |
| | Rice straw | | [32] |
| | Wheat husk | | [33] |
| | Wheat straw | Pyrolysis, 500 °C, 20 min | [34] |
| Forest products | Reed straw | Pyrolysis, 500 °C, 6 h | [37] |
| | Invasive plant | | [39] |
| Industrial by-products | Sewage sludge | | [42] |
| | Poplar sawdust | Thermal annealing temperatures, 500, 600, 700, 800 °C, 60 min | [43] |
| | Poplar woodchip | Pyrolysis, 300, 500, 700 °C, 3 h | [44] |
| | Algae residue | Pyrolysis, 450 °C, 2 h | [45] |
| Municipal solid waste | Pepper stalk | Pyrolysis, 450 °C, 2 h | [50] |
| | Municipal solid waste | Pyrolysis, 450 °C, 30 min | [50] |
| | Plastic | Pyrolysis, 300 °C, 12 h | [52] |
| Non-traditional materials | Bacteria | Pyrolysis, 800 °C, 2 h | [53] |
| | Vermicompost | Pyrolysis, 500 °C, 2 h | [55] |

*2.5. Biomass Feedstock Processing Technology*

There are many types of biomass feedstock treatment technologies, mainly classified as pyrolysis, gasification, hydrothermal carbonisation, and hydrothermal carbonisation [59,60]. Among them, pyrolysis technology is the most common, and can be divided into slow pyrolysis and fast pyrolysis according to the pyrolysis temperature and pyrolysis time. Among them, microwave pyrolysis is considered to be one of the most advanced technologies. It is worth mentioning that hydrothermal carbonisation is also a commonly used technology for wet biomass waste treatment technology, which overcomes the shortcomings of traditional thermochemistry. However, both biomass chars prepared using hydrothermal carbonisation technology show different applications, with orange peel-derived nano-biochar being used for targeted cancer therapy and olive tree-derived biochar exhibiting antimicrobial properties. This is due to the different feedstocks of the two resulting in differences in their functional groups, chemical bonding and hence chemical and physical properties [60,61]. This also shows that the treatment technology for the same biomass feedstock can be individual depending on the feedstock.

## 3. Biochar Modification Technology

Many methods have been used to modify or enhance the properties of biochar. There are two main categories, the first of which is specific to the object of the modification and can be divided into the modification of biomass feedstock and that of biochar. The second category is based on the method of modification and is divided into physical, chemical and biological methods, and the use of biochar as a substrate in combination with various substances. In the following, the different modification methods are described and compared with each other. The methods for modifying biochar are summarised in Figure 2.

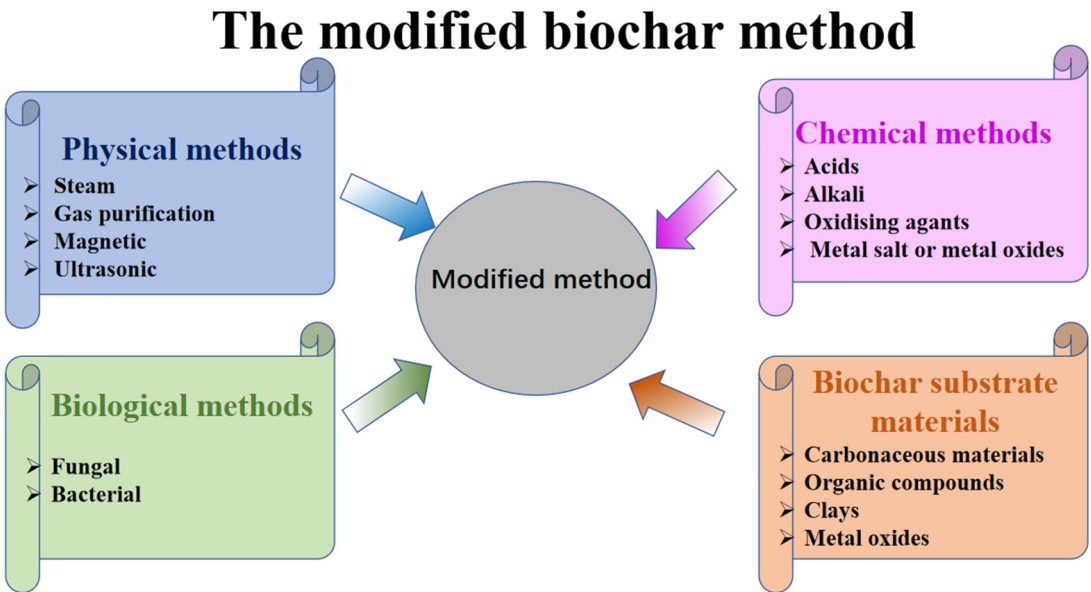

**Figure 2.** The biochar modification methods.

### 3.1. Physical Methods

Physical methods include steam, gas purification, magnetic and ultrasonic methods. Steam activation has had a positive effect, increasing the surface area of the biochar without affecting other properties. Some researchers have used controlled tests of adsorption experiments on steam-activated and non-steam-activated biochar to illustrate the alterations in biochar via steam activation. From this, it was concluded that steam activation improves the capacity for Pb(II) adsorption by enhancing the surface area of the biochar. Gas purification is similar to vapour modification in that they both enhance adsorption capacity. There are also two gas options for gas purification: carbon dioxide and ammonia. Both gases can enhance adsorption, but the specific changes they bring to biochar are different. The two gases brought about different effects, with a significant increase in the surface area of the biochar through $CO_2$ modification, and a significant enrichment in nitrogenous compounds on the surface of the biochar through $NH_3$ modification [62]. The magnetic treatment differs from the first two in that the magnetic treatment focuses on the subsequent recycling process. There are experiments using magnetic biochar to remove ciprofloxacin from aqueous solutions, and it can be separated by using magnets [63]. The final modification technique is ultrasonication, which also improves the adsorption performance of the biochar. Ultrasonic activation reduces the ash content of the biochar, promotes the formation of pore channels and increases the specific surface area [64].

### 3.2. Chemical Methods

Chemical modification is a widely used modification method. It mainly includes acid, alkali, oxidant, metal salt and metal oxide modification.

#### 3.2.1. Acids and Oxidising Agents

Acid-modified biochar brings positive changes to its physicochemical properties. Common acid-modified solvents include hydrochloric acid, sulphuric acid and phosphoric acid, among others. Reed-derived biochar was modified using hydrochloric acid, and the adsorption capacity of the treated biochar was higher than that of the original biochar [65]. The use of phosphoric acid-treated biochar can easily remove pesticides from water after treatment [66]. Although both of these are acid treatments, the surface area of the biochar modified by the different acid solutions varies: with hydrochloric acid modification, surface area increases, while with phosphoric acid treatment, surface area decreases. Due to the high cost of acid treatment and the need to guard against other environmental problems that

may arise, research is being conducted into the use of oxidants to modify biochar, providing an alternative research direction. As an example, hydrogen peroxide modification enhances the ability of biochar to remove heavy metals from water bodies [67].

### 3.2.2. Alkali Modification

The main changes brought about by alkaline modification are an increase in surface area and an increase in oxygen-containing functional groups. Commonly used alkali agents include potassium hydroxide and sodium hydroxide [2]. This is due to the abovementioned changes: that the potassium-hydroxide-modified biochar shows a greater As(V) adsorption capacity than the original biochar [68].

### 3.2.3. Metal Salt or Metal Oxide Modification

Metal salts or metal oxides can be modified in two ways: (1) by mixing them with raw materials and then synthesising the biochar, and (2) by preparing the biochar first and then soaking the biochar with metal salts or metal oxides [2]. Some experiments have used ferric chloride and magnesium chloride to modify woody biochar, and the modified biochar has an enhanced cation exchange capacity and an increased number of oxygen-containing functional groups, resulting in the enhanced adsorption of the metals [69].

### 3.3. Biological Methods

Biological modification methods have been less widely developed than physical and chemical modification methods. However, two biomass biomodification techniques, namely, fungal decomposition and bacterial digestion, have also been studied, and their ability to improve the quality of biochar has been clarified [13]. Studies have confirmed that biomass decomposed by fungi and digested by bacteria can enhance the cadmium adsorption capacity of the derived biochar. In order to better develop biomodification, it is necessary to develop more novel and effective bacterial and fungal species. In addition to this, an in-depth understanding of the process of biomodification is needed, as well as an understanding of the main mechanistic processes. It is also important to consider whether it is possible to develop the method industrially and to carry out extensive experiments to analyse the data and assess the real risks before applying it on a large scale.

### 3.4. Biochar Substrate Materials

A final means of modification is using biochar-based materials. Biochar-based composites can be constructed using metal oxides, clays, organic compounds or carbonaceous materials such as graphene oxide or carbon nanotubes in conjunction with biochar. Firstly, biochar compounded with clay minerals such as kaolin, montmorillonite or bentonite can change the physicochemical properties of the biochar. Using vermiculite as an example of a modified mineral material, modified rice straw biochar improves its enhanced chemical bonding and stability, and more notably, the modified biochar releases less or slower carbon dioxide than unmodified biochar [70]. Experiments have also been conducted using three biomass feedstocks pretreated with montmorillonite or kaolin, namely, bamboo, bagasse and hickory woodchips. The modified clay biochar composites have an improved sorption capacity and can also be used to improve soil quality and sequester carbon [71].

Secondly, in addition to clay minerals, biochar can be combined with carbonaceous materials. Either graphite oxide-modified biomass or that modified with carbon nanotubes can be used to adsorb heavy metals [72]. The difference in the latter is that, again, using sodium dodecyl benzene sulfonate dispersion, its adsorption capacity is the strongest [73]. Biochar modified with carbonaceous materials produces a wide range of sorbents and can be used as a sustainable alternative for wastewater treatment.

In addition to the above options, biochar can also be modified with organic compounds. Just as amino-modified biochar shows good adsorption of Cu(II) [74], similarly, chitosan-modified biochar has a high lead adsorption capacity [75]. In general, the modification of

biochar with amines or coating the biochar with chitosan can improve its performance as a soil conditioner or sorbent.

Finally, biochar can also be compounded with layered bimetallic hydroxides. Laminated double hydroxide–biochar (LDH-BC) composites are now receiving increasing interest because of their use as sorbents and catalysts for the removal of pollutants. Laminated double hydroxide–biochar composites can be selected according to the needs of the researcher or the characteristics of the contaminant to be treated. LDH-BC composites have good stability, abundant surface functional groups, excellent anion exchange capacity and good electronic properties [76].

*3.5. Comparison of Different Modification Methods*

Among these four modification methods, physical modification and biological modification are the least polluting to the environment, while chemical methods using acid and alkali reagents are not only more costly, but also cause different degrees of damage to the environment and are prone to secondary damage. However, biomodification is less developed than the other three categories precisely because of its potential drawbacks. In contrast, laminated dihydroxylate–biochar composites in biochar matrix materials are available in a variety of options, providing researchers with a wide range of choices. Future experimental studies where multiple modification techniques can be combined with each other will make fuller use of the material. Table 2 shows the various methods of biochar modification and their variations. The advantages and disadvantages of different modification techniques for various biochars are listed in Table 3.

**Table 2.** Various methods of biochar modification and their variations.

| Type of Modification | Specific Method | Modified Object | Change Observed | References |
|---|---|---|---|---|
| Physical methods | Steam | Canola straw biochar | Increased surface area | [56] |
| | Gas purification | Cotton straw biochar | Increased surface area; enrichment of surface nitrogenous compounds | [62] |
| | Magnetic | Chinese herbs biochar | Magnetic | [63] |
| | Ultrasonic | Caragana korshinskii biochar | Reduced ash; increased specific surface area | [64] |
| Chemical methods | Acids and oxidising agents | Reed biochar | Reduced ash; increased surface area | [66] |
| | Acids and oxidising agents | Rice straw biochar | Increase in surface functional groups and aromatisation | [66] |
| | Acids and oxidising agents | Peanut hull biochar | Increased surface oxygen functional groups | [67] |
| | Alkali | Municipal solid waste biochar | Increased surface area; increased oxygen-containing functional groups | [68] |
| | Metal salts or metal oxides | Woody biochar | Enhanced cation exchange capacity | [69] |
| | Biological | Casuarina biomass | Grey component; specific surface area; pH value increase | [13] |
| | Biochar substrate materials | Rice straw biochar | Enhanced chemical bonding; stability | [70] |
| | Biochar substrate materials | Bamboo, bagasse and hickory wood chips | Increased adsorption capacity | [71] |

**Table 3.** Advantages and disadvantages of different modification techniques for biochar.

| Type of Modification | Specific Method | Disadvantages | Vantage | References |
|---|---|---|---|---|
| Physical methods | Steam | Affected by type of feedstock and production conditions | Increased surface area | [56] |
| | Gas purification | Temperature-dependent | Increased surface area; enrichment of surface nitrogenous compounds | [62] |
| | Magnetic | pH-affected | Cost-effective; easy to separate' | [63] |
| | Ultrasonic | Application less | Simple structure; suitable for industrial applications | [64] |
| Chemical methods | Acids and oxidising agents | Prone to secondary contamination | Increased surface oxygen functional groups | [66] |
| | Alkali | Prone to secondary contamination | Increased surface area; increased oxygen-containing functional groups | [68] |
| | Metal salts or metal oxides | Prone to secondary contamination | Inexpensive; enhanced cation exchange capacity | [69] |
| | Biological | Temperature-dependent; unstable effect | Grey component; specific surface area; pH value increase | [13] |
| | Biochar substrate materials | Temperature-dependent | Enhanced chemical bonding; stability; improved stability | [70] |

## 4. Environmental Application of Biochar

Biochar is used in a wide range of environmental applications, in the atmosphere, water and soil. This article will look at its specific application roles, such as an adsorbent, catalyst, activator and soil conditioner and in carbon sequestration.

### 4.1. Adsorbent

Biochar as an adsorbent can adsorb and treat a large number of pollutants, which can be divided into three main categories: inorganic, organic and radioactive elements.

Starting with the categories of pollutants treated via adsorption, the first is inorganic pollutants, which are mainly divided into other substances such as heavy metals and phosphates. Heavy metals are a serious and widespread pollution hazard. There are studies on the removal of heavy metals by adsorbents, mainly for stormwater systems. Experiments have shown that biochar has different adsorption efficiencies for different heavy metals, with high removal rates for Pb and Cr, and low removal rates for Ni and Cd [77]. In addition to water systems such as stormwater systems, sorbents have been studied for heavy metal removal from wastewater and other water bodies. Due to the wide range of heavy metals, eight are described below.

This section presents a comparative discussion on the adsorption effects of heavy metals of the same cycle, based on the principle of the same cycle. Firstly, for Cr, Ni and Cu in the fourth cycle, there were experiments to study the removal characteristics of Ni(II), Cr(VI) and Cu(II) in mixed wastewater via a symbiotic system of free strain L1, peanut shell biochar and the strain immobilised on it, and the results showed that the removal rate of the symbiotic system was higher than the effect of a single strain [78]. This is because the various functional groups in the co-system surface contribute to the removal of heavy metal ions. Meanwhile, there are other studies on the effect of different groups on adsorption, such as the role of O- and N-containing groups in the selective adsorption of Cr(VI), with O-containing groups making a greater contribution than N-containing groups [79]. This is followed by fifth-cycle heavy metals such as cadmium and antimony. Antimony is widely used because of its health risk, and Sb(III) is adsorbed using biochar or modified biochar (biochar loaded with chitosan); the maximum adsorption capacity of the modified biochar is 16 times that of the original biochar [80]. In addition to the abovementioned means of modification, Sb(V) can be adsorbed on phosphogypsum-modified biochar, which has a

maximum adsorption capacity more than twice that of the original biochar. The mechanism of Sb(V) removal by using phosphogypsum-modified biochar showed that the chemical composition of the modified biochar was carbon, calcium, sulphur and oxygen based on the experimental characterisation. Compared to the modified biochar before and after adsorption, the characterisation showed that agglomerated particles with calcium and oxygen as the main chemical components dominated the adsorption of antimonate [81]. Cadmium, which has the same cycle as antimony, can also be adsorbed by modified biochar. The oxidant-modified biochar of *Platycodon grandiflorus* leaves was twice as effective in adsorbing cadmium from aqueous solutions as the original biochar. There are many options for oxidation means, among which $KMnO_4$-modified biochar (MBC) can be more effective in removing $Cd^{2+}$ from aqueous solutions. This is due to the fact that MBC has a large specific surface area with many particles (manganese oxide) distributed on the surface, where complexation with manganese oxide is the main mechanism. In addition, the complexation of oxygen-containing groups, precipitation, cation–$\pi$ traction and ion exchange also provides contributions to adsorption [82]. As above, the adsorption of Cd(II) using the modified biochar (phosphate (P)-modified) was also superior to that of the original biochar. The means of modification vary and so do the principles. Unlike the previous two, the adsorption principle of orthophosphate-modified biochar is due to the formation of a variety of different iso-Cd-P precipitates, which enhances the adsorption capacity of Cd. In addition, the higher cation exchange efficiency among the modified biochar is key to improving its ability to adsorb [83]. Finally, there are the heavy metals of the sixth cycle: mercury, thallium and lead. Mercury is not uncommon in our daily lives. The removal of elemental mercury from coal combustion flue gases was studied using P-doped biochar and it showed good adsorption properties [84]. While thallium has been experimentally prepared, programmable synthesised exfoliated biochar nanosheets have efficient selective adsorption of thallium [85]. The same S-rich biochar and magnetic biochar also have outstanding adsorption capacity to adsorb the heavy metal ion Pb(II) [86,87]. Of course, in addition to the abovementioned biochar adsorbents for the efficient and selective adsorption of one heavy metal, there are also biochar adsorbents for the simultaneous adsorption of several heavy metals. Additionally, biochar that can adsorb several heavy metals is not unique. For example, there are various types of biochar that can adsorb both lead and cadmium. The adsorption of Pb(II) and Cd(II) in water can be achieved with $KHCO_3$-activated biochar loading MgO, magnesium amidoferrate–biochar composites, 3D-layered porous biochar, and sulphonated biochar, which all have high adsorption capacity [88,89]. To summarise, the adsorption mechanisms in the above experiments include diffusion, electrostatic attraction, ion exchange, co-precipitation, complexation, pore filling, chemisorption, strong binding effect of sulphite functional groups, oxygen functional group complexation, physical adsorption, high electronegativity of molecules and pH value of biochar [90,91]. Among them, the combined effect of the adsorption of two heavy metals by $KHCO_3$-activated biochar loaded with MgO was the most efficient. Biochar adsorbents not only adsorb heavy metals in water and fumes, but also immobilise and fix heavy metals in soil [92]. This will not be repeated here and will be discussed again in Soil Amendments. Table 4 shows the removal of heavy metals from the environment with biochar and their specific values.

**Table 4.** Removal of heavy metals from the environment with biochar.

| Heavy Metal | Initial Concentration | Material | Modification Method | Maximum Adsorption Capacity | Removal Efficiency | Preparation Method | References |
|---|---|---|---|---|---|---|---|
| Ni(II) | 20 mg/L | Peanut shell | Load strains | | 81.17% | 500 °C for 2 h | [78] |
| Cu(II) | 20 mg/L | Peanut shell | Load strains | | 45.84% | 500 °C for 2 h | [78] |
| Cr(VI) | 20 mg/L | Peanut shell | Load strains | | 38.21% | 500 °C for 2 h | [78] |
| Sb(III) | 40 mg/L | Pteridium aquilinum | Loading chitosan | 168 mg/g | 88% | 80 °C for 24 h | [80] |
| Sb(V) | 24.36 mg/L | Grain | Phosphogypsum modification | 8123 mg/kg | | 300 °C; 400 °C; 500 °C | [81] |
| Cd(II) | 50 mg/L | Eastern maple leaf | Oxidiser modification | 52.5 mg/g | 98.57% | 400 °C for 4 h | [82] |
| Cd(II) | 50 mg/L | Apple branch | Phosphate modification | 116 mg/g | 99.98% | 500 °C for 2 h | [83] |
| T1(I) | | | | 382.38 mg/g | 90% | 900 °C | [85] |
| Pb(II) | | Wheat straw Corncob | Sulphur modification | 421.8 mg/g | | 600 °C for 3 h | [86] |
| Pb(II) | | Wheat straw | Magnetisation | 817.64 mg/g | | | [87] |
| Pb(II) | 600 mg/L | Peanut shell | KHCO₃ activation and MgO nanoparticles incorporation | 1625.5 mg/g | | 400 °C for 2 h | [88] |
| Cd(II) | 100 mg/L | Peanut shell | KHCO₄ activation and MgO nanoparticles incorporation | 480.8 mg/g | | 400 °C for 2 h | [88] |
| Cd(II) | 500 mg/L | Rice husk biochar | Introduction of specific functional | 195.5 mg/g | | 450 °C | [89] |
| Pb(II) | 500 mg/L | Rice husk biochar | Introduction of specific functional groups | 198.93 mg/g | | 450 °C | [89] |
| Pb(II) | | Coconut shell | In situ formation of zinc oxide templates | | | | [90] |
| Cd(II) | | Coconut shell | In situ formation of zinc oxide templates | | | | [90] |
| Pb(II) | 200 mg/L | Laminated wood | Sulfonation | 191.07 mg/g | | 180 °C for 0.5 h | [91] |
| Cd(II) | 100 mg/L | Laminated wood | Sulfonation | 85.76 mg/g | | 180 °C for 0.5 h | [91] |

Biochar also adsorbs phosphate pollutants well, but it has its drawbacks. There are various options for biochar for the removal of phosphate contaminants from water, for example, seawater-modified biochar, metal/metal oxide biochar, calcium alginate–biochar composites, biochar in dewatered dry sludge and many others [93–96]. Overall, the best adsorption was achieved using seawater-modified biochar, with a maximum adsorption capacity of 181.07 mg/g [93]. The mechanisms behind the adsorption involved electrostatic attraction, ligand exchange, precipitation and internal sphere complexation. Relatively less effective were the metals/metal oxides on biochar, with a maximum sorption of 9.75–25.19 mg/g, where the metal oxides played an important role in terms of sorption and co-precipitation [94]. In summary, there are many mechanisms of adsorption, such as electrostatic attraction, chelation, surface complexation, ion exchange, co-precipitation and complexation, among other effects. However, these are all related to the properties of the biochar itself and the different means of modification.

There has also been much research into the removal of organic pollutants by sorbents, which can be classified into pharmaceutical and printing categories according to the pollutant classification. Pharmaceuticals include not only antibiotics for humans and animals, but also insecticides and herbicides. The widespread use of antibiotics in people's daily lives or in livestock and seafood farming has also led to a number of environmental problems, for which the use of biochar adsorption is a good solution. In this paper, 4 of the 18 top-priority antibiotics (including macrolides and other categories such as quinolones, sulphonamides and tetracyclines) selected from the decennial national screening dataset are selected for discussion and study [97]. Firstly, the quinolones are exemplified by enrofloxacin, and the removal efficiency of enrofloxacin can be improved by using biochar prepared from Fe- or Zr-modified marine agricultural solid waste. This is because this significantly improves the surface properties of the biochar [98]. In addition to the above biochar, alkali- and bimetallic salt-modified sludge biochar was used for the efficient adsorption of fluoroquinolone antibiotics (ciprofloxacin, norfloxacin and ofloxacin) in water. It had a maximum adsorption capacity of 49.9, 55.7 and 47.4 mg/g, respectively, and showed good magnetic properties and stability [99]. Next to sulphonamides, for sulphamethoxazole algae-derived biochar, boric acid-activated biochar and $FeCl_3$-activated bermudagrass-derived biochar can be used, all of which have good adsorption properties [100–102]. For example, algal biochar showed excellent adsorption capacity for SMX (4874–4879 mg/kg). Once again, they are tetracyclines, and both hydrothermal mesoporous biochar and bog leaf zinc-containing biochar can be used to adsorb tetracyclines [103,104]. The adsorption mechanisms for both include surface complexation, $\pi$–$\pi$ interactions, hydrogen bonding, pore filling and n–$\pi$ interactions [105]. Pesticides and herbicides also have an impact on the environment. Some studies have used agricultural waste biochar to adsorb leuconazole, or N-doped biochar to adsorb atrazine, both of which have high adsorption capacity [106,107].

In addition to the above organic pollutants, there are also dyes and volatile organic compounds that can be removed using biochar. The use of modified rice husk and EDTA/chitosan bifunctional magnetic bamboo biochar for methyl violet, malachite green and methyl orange was effective [107,108]. Volatile organic compounds were also enhanced with different modification techniques: ammonia or hydrogen peroxide ball-milled biochar for the adsorption of volatile organic compounds [109].

Both modified horse manure biochar and modified pig manure biochar can be used for the adsorption of uranium (VI). Although both are made from animal dung, they have different adsorption effects on uranium (VI). The maximum adsorption capacity of the former is 516.5 mg/g, while the maximum adsorption capacity of the latter is 979.3 mg/g [110,111]. This is due to the different modifications chosen, even for the same biochar feedstock, and the different adsorption effects of the different modifications. Table 5 shows the use of biochar to remove antibiotics from the environment.

**Table 5.** Removing antibiotics from the environment using biochar.

| Absorbent | Biochar | Maximum Adsorption Capacity | References |
|---|---|---|---|
| Fluoroquinolone antibiotics | Sludge biochar modified with alkali and bimetallic salts | 55.7 mg/g | [99] |
| Sulphamethoxazole | Algae-derived biochar | | [100] |
| Sulphamethoxazole | Boric acid-activated biochar | | [101] |
| Sulphamethoxazole | $FeCl_3$-activated bermudagrass-derived biochar | | [102] |
| Tetracyclines | Hydrothermal mesoporous biochar | | [103] |

*4.2. Catalyst and Activator*

Biochar can be a catalyst or an activator. Firstly, as a catalyst, it can degrade antibiotics and dyeing pigments in line with adsorbents [33], for instance, the adsorption and photocatalytic degradation of methylene blue using ZnO and biochar composites. Firstly, adsorption plays a major role in MB removal, and secondly, photocatalysis also degrades adsorption through the generated free radicals. Additionally, adsorption may be greatly affected by electrostatic attraction, and its photocatalytic mechanism is mainly provided by ZnO, which is a semiconductor used for photocatalysis. It is well known that photocatalysis technology is currently facing two major challenges. One is the rapid compounding of photogenerated electron–hole pairs, and the other is the limitation of photocatalysts for the spectral absorption range. In this study, the electrons are transferred from the valence band of ZnO on the surface to the electrons in the conduction band, and biochar has a role in promoting the reaction. The active substances generated in this process are superoxide radicals and hydroxyl radicals, which are the result of the reaction between electrons and $O_2$, respectively, and holes may also react with water. Biochar also has some photocatalytic ability to form heterojunctions with zinc oxide to generate the abovementioned active substances more effectively. In conclusion, the new ball-milled ZnO–biochar nanocomposites have good photocatalytic carriers, successfully loaded ZnO particles and improved the complexation phenomenon between holes and photo-generated electrons. The Figure 3 below shows the mechanism for the removal of methyl bromide from the composite material under light irradiation [112]. Similarly, researchers can choose from a variety of modifications to catalyse degradation, such as combinations with metal oxides, magnetic materials and Bi oxides, all of which can be used to degrade substances [6,38]. Unlike other applications, biochar can also be an activator, activating persulphates to degrade pollutants. The activation of persulphate systems has become a major branch of advanced oxidation processes, and biochar is also considered a potential material in this field.

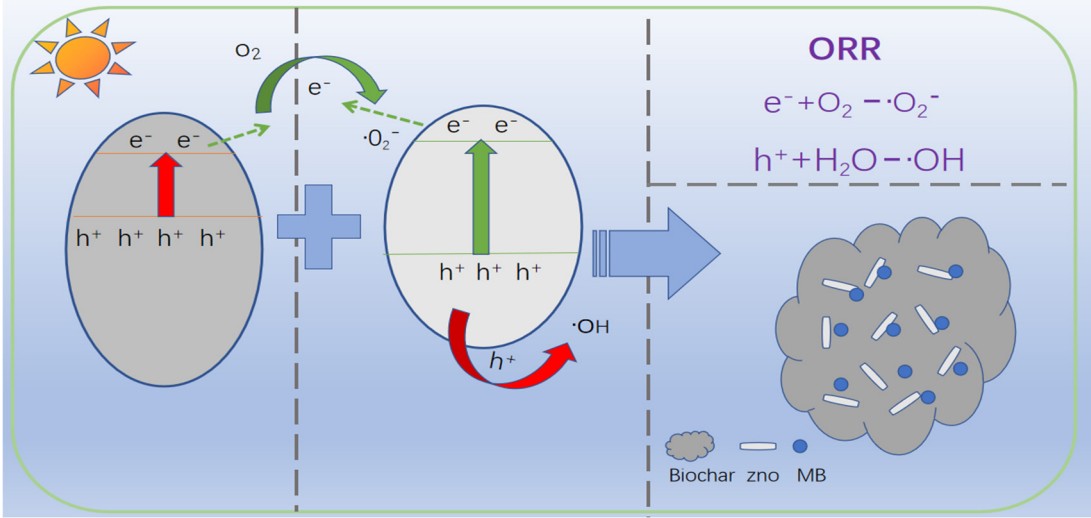

**Figure 3.** Mechanisms for the removal of methyl bromide using light.

Biochar is now receiving a lot of attention and research as a persulfate activator and photocatalyst. Both of these uses are described below. The primary focus is on biochar as a persulphate activator. In particular, its use in the removal of degraded organic matter after activation has been much studied, showing excellent performance in the treatment of organics with benzene rings. Biochar-loaded iron sulphide is used as an activator of peroxynitrite (PS) for the removal of benzene from water and shows excellent perfor–mance in terms of adsorption and subsequent degradation [113]. Other studies have synthesised $\alpha$–FeOOH–biochar as an activator for the degradation of phenol using PS perthi –olate, and it has also shown good catalytic oxidation properties [114]. In this experiment, $\alpha$–FeOOH–biochar composites were synthesised via a combination of ball milling and liquid deposition phase techniques and applied to the activation of PS used to degrade phenol. From a mechanistic point of view, the functional groups, the dissolved organic matter in the biochar, the loading of Fe elements and the higher graphitisation and defective structure all favour the activation of PS to form radicals for the oxidation of phenol. Among them, sulphate radicals and hydroxyl radicals play a dominant role in the oxidation of phenol. Both showed excellent performance, but their specific activation produced significantly different radicals, with the former activating radicals (sulphate radicals, hydroxyl radicals) and the latter activating PS to form radicals (sulphate radicals, hydroxyl radicals, superoxide radicals and others). In addition to this, it can also be used to degrade antibiotics. In this paper, two antibiotics are selected for comparative discussion, one of which is soybean straw biochar as an activator of PS for tetracycline degradation and disinfection of tetracycline-resistant *E. coli* [115]. The second is nitrogen-doped sludge-derived biochar for the activation of sodium persulfate (PMS) for the degradation of sulphamethoxazole [116]. While the former degradation is mainly through surface free radicals and sterilisation attributed to sulphate radicals, the latter nitrogen doping can improve the catalytic activity of biochar. Whereas the four types of biochar mentioned above all degrade a specific pollutant singularly by activation, the magnetic 2D/2D oxygen-doped g-$C_3N_4$–biochar composite activates persulfate to degrade not only sulphamethoxazole, but also phenol, atrazine, nitrobenzene and carbamazepine [117]. This shows that such composites exhibit a wide range of applicability. Then, there are organic pollutants such as bisphenol A and its analogues, which are also treated well. Fe–N co-doped biochar is used as a peroxynitrite activator to degrade bisphenol A [118]. Bisphenol S, a BPA analogue, can be degraded by composites, combining magnetic spinel and sludge-derived biochar [119]. Biochar is mentioned in the above article as being able to act as an adsorbent and also as an activator of it, and some researchers have combined these two properties in one biochar. The selenium–nitrogen co-doped biochar has become a bifunctional catalyst for adsorption–oxidation, not only for phenol but also for the activation of persulfate [120]. In summary, the activation of persulphate by biochar can degrade a wide range of pollutants, such as antibiotics, printing and dyeing and other organics. However, there are still parts of this that need to be investigated, and highly toxic intermediates may be produced during the degradation process, which can affect their application in practice [121]. In addition, a more detailed understanding of the degradation process and mechanisms is needed to help reduce the production of highly toxic intermediates. There is, therefore, a need to construct more rational biochar-based catalysts and to increase the depth of research into activation mechanisms [122]. Different biochar activation of persulfate produces different active substances, and the Figure 4 below suggests possible activation mechanisms.

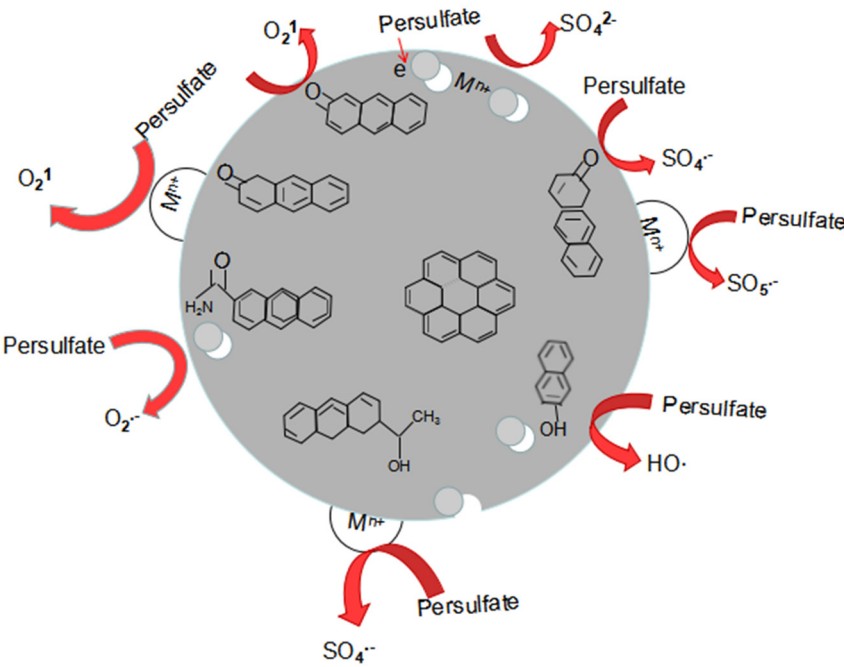

**Figure 4.** Biochar-activated persulfate mechanisms.

Biochar as an activator has been discussed above, and the following discussion will focus on biochar as a photocatalyst. Photocatalysts prepared using biochar can be one of the options for the removal of organic pollutants. Table 6 summarises the information on biochar when used as a photocatalyst in the environment. The following section focuses on organic pollutants present in pharmaceuticals and medicines. Firstly, with regard to organic pesticides, biochar–$\alpha$–$Fe_2O_3$–magnesium oxide composites can photocatalytically degrade organophosphorus pesticides and enable the recovery of orthophosphates [123]. In addition to organic pesticides, there is also estrone, which has been studied for photocatalytic degradation using bismuth-containing photocatalytic biochar, and which has excellent performance compared to virgin biochar [31]. The mechanism of its biochar–$\alpha$–$Fe_2O_3$–MgO composite is mainly UV light activating persistent radicals on the surface of the above composite to provide electrons for $O_2$, which generates superoxide radicals and then hydroxyl radicals. Additionally, in the process of photocatalytic degradation holes, superoxide radicals and hydroxyl radicals are the main causes of degradation. In addition to the above studies, the use of biochar photocatalysts for the removal of antibi–otic–like contaminants has received widespread attention. Green photocatalysts $Fe_3O_4$/Bi–OBr/CQDs derived from maize cob biomass can be used for the degradation of carbamazepine [124]. There are more studies using biochar for the photocatalytic degra–dation of antibiotics, such as $Fe_3O_4$–BiOBr–BC photocatalysts for the photodegradation of carbamazepine and titanium dioxide stacked on reed straw biochar for sulphamethox–azole under visible light LED irradiation [125,126]. The hydroxyl radicals in the active material produced by the bismuth-containing photocatalytic biochar (BiPB) composite prepared in the above study are the main reason for its good photocatalytic performance. In addition to the addition of bismuth, which is beneficial to control the persistent radicals, the biochar can also have a graded porous structure. Furthermore, the addition of biochar also improves the separation and transfer efficiency of charge carriers. Under neutral pH conditions, it is more favourable for the composites to generate hydroxyl radicals and thus degrade estrone. Both showed good photocatalytic activity and the mechanism was similar for both. The specific mechanism is shown in the Figure 5 below, where X is either $TiO_2$ or BiOBr and Y is $Fe_3O_4$ or ZnO.

**Table 6.** Biochar in the environment as a photocatalyst.

| Degradable | Material | Photocatalyst | Degradation Rate | References |
|---|---|---|---|---|
| Methylene blue | Bamboo | ZnO/biochar | 95.19% | [112] |
| Organic phosphorus pesticides | Rice straw | biochar/$\alpha$–$Fe_2O_3$/MgO | 82.30% | [123] |
| Carbamazepine | Corncob | $Fe_3O_4$/BiOBr/CQDs | 99.52% | [124] |
| Carbamazepine | Reed straw | $Fe_3O_4$/BiOBr/BC | 51.50% | [125] |
| Sulphamethoxazole | Reed straw | Zn-$TiO_2$/pBC | 81.21% | [126] |

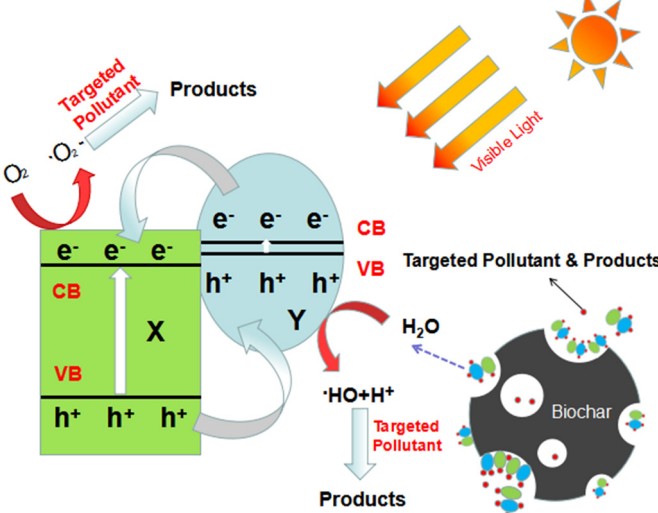

**Figure 5.** Biochar photocatalytic mechanism.

In addition to the above, common photocatalysts combining biomass charcoal and $TiO_2$, there are nanoparticles composited with them to make new hybrid nanomaterials. This is a combination of the advantages of both: the composite will have the ability to exhibit good surface properties, high porosity and stable chemical properties and will have higher photocatalytic capacity [127]. However, such composites may indirectly release toxic substances. Therefore, biodegradable and environmentally non-toxic materials should be selected for specific applications in the future [128]. Biochar-based photocatalytic materials have great potential for water treatment and for future commercialisation in industry, but considerable efforts are still needed to reach industrial-scale requirements [129]. Examples include technical feasibility, recyclability of materials, toxicity testing of intermediate by-products, economic practicality and safety for the environment. Overall, the implementation and utilisation of such photocatalysts will have to be studied for a long time.

*4.3. Soil Remediation and Improvement*

Biochar can also be used as a soil remediator and amendment. The applications of biochar in soil are diverse, with its ability to remove organic and inorganic contaminants from the soil, neutralise soil pH and improve soil fertility. For instance, biochar materials combining ball milling and phosphorus can effectively adsorb Cd and Pb, thus reducing the mobility of heavy metals in the soil and mitigating heavy metal stress on maize plants [130]. In this study, it was shown that biochar in soil enhances the sorption of Cd and Pb through electrostatic effects, surface precipitation and complexation. The mechanism of heavy metal adsorption in water is somewhat similar to that of biochar, where heavy metals are adsorbed in water through electrostatic, chelation, surface complexation, hydrogen bonding, ion exchange, co-precipitation, complexation and so forth. In addition to the abovementioned biochar, which is only used to remove heavy metals, there are polyacrylic acid-grafted chitosan and biochar composites that can be used as soil conditioners, not only to reduce the amount of heavy metals in the soil, but also to improve the nitrogen cycle and change the structure and function of the soil microbial community [131]. After remediation

of the environment, the biochar material and the adsorbed metals can be recycled and reused, which is of resource reuse importance [132]. In order to expand the commercial use of biochar, various factors such as technical production, environmental impact and economics need to be considered; additionally, the advantages and disadvantages of various applications of biochar in soil are discussed in depth [133].

Similarly, most biochar is alkaline and can raise the pH of acidic soils. Taking advantage of this principle, biochar is therefore used as a soil conditioner to raise the pH of acidic and slightly acidic rice soils, improving the use and uptake of Si by rice and thus promoting rice growth [134]. Biochar not only promotes crop growth, but also improves soil fertility. The addition of paper mill-derived biochar increases the biomass of radish, but not of that wheat or soybeans [2]. This therefore shows that the same biochar has different effects on different crops. For tobacco, a special plant with a high content of heavy metals, there is also research into how biochar can be used to solve this problem: biochar derived from tobacco stalks can be used in tobacco-grown and contaminated soils, where it can convert heavy metals into a less harmful form, which can be used to remediate the soil and improve tobacco productivity [135]. Although the application of biochar in soils has many advantages, its long-term effects on soils are still unclear. Therefore, in order to better use biochar and reduce possible risks, more research should be focused on the long-term effects of biochar on soils and risk assessment of practical applications in response to the results.

### 4.4. Carbon Sequestration

The massive use of fossil fuels by humans leads to massive emissions of carbon dioxide that cause global climate change and pose a major environmental hazard [136–138]. Climate change has become a worldwide concern, and research is being carried out on how to reduce $CO_2$ emissions in the atmosphere. Biochar is not only a soil amendment but also a climate change mitigation tool. Carbon sequestration has been suggested as a way to mitigate $CO_2$ emissions from soils.

Many studies have been carried out; however, the effect of biochar on soil carbon sequestration has not provided consistent results. Biochar reduces C mineralisation, leading to greater C sequestration, and this effect initially increases and then decreases and stabilises over time [139]. In summary, the effect of biochar on carbon sequestration is unclear. Its effect varies with the raw material and preparation conditions [2]. This is followed by specific experiments to illustrate the concrete effects of carbon sequestration with biochar. One experiment used two years of field trials where the addition of BC to dairy manure DM1 at high N concentrations, DM2 at low N concentrations and inorganic N resulted in varying degrees of reduction in $CO_2$, $CH_4$ and $N_2O$ emissions [140]. Additionally, both two-year experiments had researchers using different biochar, i.e., maize stover biochar amendment, on greenhouse gases in sandy loam soils, with an incremental effect of total $CO_2$ emission reduction in both growing seasons [141]. In other studies, however, very different conclusions were reached, with no significant effect of biochar on cumulative soil $CO_2$ emissions in the laboratory or in field-scale experiments [142]. Other studies also used biochar to form an organic carrier to prepare a good $CO_2$ adsorbent with an adsorption capacity of 4.23 mmol $CO_2$ $g^{-1}$ and its performance remained stable after 30 cycles of adsorption experiments [143]. These reports show that a large number of experiments and pilot-scale studies are still needed to investigate and discuss the mechanisms of carbon sequestration in detail.

### 4.5. Other Effects

In addition to the abovementioned environmental applications of biochar, there are other applications for biochar. Most studies have shown that in the field of environmental remediation and supercapacitors, biochar morphology has an influence on its performance [144]. Biochar is being explored for applications in construction materials, due to its ability to sequester carbon and thus improve the performance of construction materials [17]. Biochar can also be used in fuel cells. Other applications of biochar are specified below; first

of all, biochar as a building material. Nowadays, biochar is added to cement composites as nanoparticles, making it suitable for use as a filler in cement materials [145]. The use of biochar-containing building materials can also capture carbon dioxide from buildings and their structures, which can reduce greenhouse gas emissions by a further quarter [146].

The second is the impact of biochar on fuel cells. Biochar is a promising fuel for direct carbon fuel cells, using pure walnut shell biochar and iron-laden biochar for a fuel cell or walnut and almond shell biochar for a direct carbon fuel cell [147–149]. Additionally, biochar can be used as a simple fuel from a wide range of sources such as ground coffee biochar, biochar derived from rice husks, biochar derived from food waste and biochar derived from sewage sludge [150–153]. Furthermore, the optimisation of process parameters can improve the quality of biochar to varying degrees. In addition to this, biochar can be used as a fuel, as a catalyst for fuel cells and as a cathode electrocatalyst support for fuel cells [154–156].

In addition to this, there are slow-release biochar compounds that have a payback period of only 4 years, which are available as commercial products in China and are comparable in price to chemical fertilisers [157]. Additionally, biochar catalysts reduce the cost of the biodiesel production process by reducing time and energy consumption [158]. There is another promising application—the use of biochar in supercapacitors. Biochar can be used as an electrode for supercapacitors and the capacitance of the electrode can be improved by appropriate modification of the biochar [159].

Biochar plays a key role in soil, water or air, i.e., the armour of the environment [160]. Specific applications are as follows: biochar not only removes heavy metals, organic matter and nanoplastics from water, but it also mitigates the spread of antibiotic resistance genes in soil. It can also be used to recover nutrients from wastewater as fertiliser. Biochar also mitigates nitrous oxide emissions during microbial denitrification [161–166]. All in all, biochar has great potential for environmental applications.

## 5. Conclusions and Perspectives

This article provides an overview of biochar, particularly in terms of feedstock, modification and environmental applications. The conversion of waste into biochar provides a practical application for the reuse of waste resources. In addition, the diverse environmental applications of biochar promote its development. Biochar is available from a wide range of sources, but its physicochemical properties are still deficient. Researchers have often used modifications to improve its physicochemical properties. These modifications can be divided into physical, chemical and biological methods and biochar matrix materials. Of these, chemical methods are the most used, with their ability to increase surface functional groups and increase surface area. However, they can be environmentally damaging. In contrast, composite materials with biochar as the matrix material have a wide range of options, providing researchers with a wide range of choices to deal with different contaminants. Biochar has a wide range of applications in the environment, treating pollutants in the atmosphere, wastewater and soil. The abovementioned articles classify them as adsorbents, catalytic activators, soil conditioners and carbon sequestration. In summary, the application of biochar has a broad future. However, there are also problems and scientific and technological barriers to future development. In particular, the following issues need to be studied and paid attention to.

(1) Developing more types of waste to prepare new biochar.
(2) Investigating combinations of modification methods to improve the performance of biochar.
(3) Investigating how biochar can be harmlessly treated or reused for disposal after adsorption saturation.
(4) Clarifying the activation mechanism of biochar.
(5) Enhancing the research and development of the corresponding photocatalytic desorption technology of biochar photocatalyst.

(6)　　Studying and elaborating the safety of biochar or modified biochar in soil and its effects on plants and animals, corresponding ecosystems and human beings.

(7)　　Conducting experiments at different scales to assess the feasibility of biochar in wastewater treatment.

(8)　　From the point of view of future development and application, more attention should be paid to the large-scale development of biochar, i.e., the selection of raw materials, preparation processes and modification means should be in line with the direction of large-scale, economic, safe and environmentally friendly development.

**Author Contributions:** Conceptualization, W.W. and J.H.; methodology, W.W.; software, W.W.; validation, W.W., J.H. and T.W.; formal analysis, W.W.; investigation, W.W.; resources, W.W. and T.W.; data curation, W.W.; writing—original draft preparation, W.W.; writing—review and editing, W.W., X.R. and X.Z.; visualization, X.R. and X.Z.; supervision, X.R. and X.Z.; project administration, X.R. and X.Z.; funding acquisition, X.R. and X.Z. All authors have read and agreed to the published version of the manuscript.

**Funding:** This project was supported by grants from the Natural Science Foundation of Jilin Province (20230101220JC), National Natural Science Foundation of China (52270038), Natural Science Foundation of Jilin Province (20220101073JC) and Province Department of Education Scientific and Technological Research Projects of Jilin, China (JJKH20220450KJ and JJKH20220447KJ).

**Data Availability Statement:** Data in support of the reported results can be found at references.

**Conflicts of Interest:** The authors declare no conflict of interest.

## Abbreviations

| | |
|---|---|
| BC | biochar |
| LDH | layered double hydroxide |
| PMS | peroxymethane sulphate |
| MSW | municipal solid waste |
| VOCs | volatile organic compounds |
| PS | peroxynitrite |

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
