# Peer review of "Research on the Preparation of Biochar from Waste and Its Application in Environmental Remediation"

_water, doi:10.3390/w15193387_

Round 1
Reviewer 1 Report (New Reviewer)
This paper provides an overview of the types of waste that can be used for biochar preparation and their specific substances, as well as summarizing methods to enhance or improve the performance of biochar, such as physical, chemical, biological and other methods.
Below are my comments.
1. Improving all figures.
2. The works chosen and reviewed for the topic in question must depend on the authors' discretion. I realize that the authors cannot cite all works related to the topic under review. However, I believe that works that report relevant techniques or results obtained in recent years for the first time should be reviewed. For example, the technique of hydrothermal carbonization has been omitted, which is a promising treatment technique for wet lignocellulosic biomass waste, as it overcomes the drawbacks of conventional thermochemical processes that require the use of dry raw materials. A paragraph is sufficient to highlight the commonalities and differences (chemistry and physical properties) between the materials under study and other products made from waste materials such as orange peels (e.g. https://doi.org/10.3390/pharmaceutics14102249) or solid waste cultivars of the olive tree (e.g. https://doi.org/10.3390/nano12050885), and explain why they have different applications.
3.The cases mentioned in this review have not been analyzed in full and should be analyzed from the point of view of preparation method, structure, performance, and other aspects. Please add.
4. Correct the numerous typos in the manuscript.
5.Uniform the text in the tables.
6. I recommend adding a list of acronyms at the end of the manuscript.
Moderate editing of English language required
Author Response
Thank you for your useful suggestions and reviewers' and the editors comments on our manuscript. We have modified the manuscript accordingly, and detailed corrections are in red in the revised manuscript, the answers to reviewers' and editor s' questions are listed below point by point:

Reviewer 2 Report (New Reviewer)
The aim of this paper is to present a comprehensive review on application of biochar absorbent in environmental remediation. The paper covers various methods developed for biochar preparation and modification. The role of biochar in environmental applications have been also well reviewed and potential mechanisms involved in different roles have been also presented. The paper contains useful information in the field of study and can contribute in the current knowledge of the issue. So, the paper may be considered for publication with Water after some modifications in below:
1. Words picked for keyword list are suitable. However, you can add some more related ones to increase the visibility of your valuable paper.
2. Please add a table to compare technical aspects of different feedstocks reviewed in your paper.
3. A CRITICAL SUGGESTION: You have reviewed different modification techniques can be used for biochar preparation. However, it is necessary to compare them from technical viewpoint. So, I may suggest you to add a table in which you can compare their positives and negatives, with a logical comparison discussion in the text body.
4. Please complete the third column in Table 4.
5. Please separation section 4. Conclusions and perspectives to two different sections of Future perspectives and Conclusions. In “Future perspectives” you can discuss about the current shortcomings and scientific and technical gaps in this field and then you can open up new horizons for future research studies.
Good luck,
Minor editing of English language is required.
Author Response
Thank you for your useful suggestions and reviewers' and the editors comments on our manuscript. We have modified the manuscript accordingly, and detailed corrections are in red in the revised manuscript, the answers to reviewers' and editor s' questions are listed below point by point:

Reviewer 3 Report (New Reviewer)
In the manuscript entitled “Research on the preparation of biochar from waste and its application in the environmental field”, the author had presented an attractive work regarding to the preparation of waste derived biochar, the relevant modification approaches, and the potential and practical utilization of such materials in environmental remediation. However, I think the manuscript still has some issues which must be improved. My comments were listed as follows
Major comments
(1) The title is lack of concise, and the concept of “environmental field” seem too wide. Better to use environmental remediation instead.
(2) The graphical abstract (topic map) need to be revised. Modified biochar should be changed to modification methods. In addition, here, what’s the connection between “Environmental application of biochar” and “Removal of pollutants”. Need to consider the logical issue of this figure.
(3) The outline of Introduction is not clear. The first paragraph needs to state the background caused by the booming generation of various types of solid waste. Followed by a brief information of biochar, and concluded with viewpoint that converting these waste into biochar materials is a promising approach to solve the issues caused by waste generation. The second paragraph seems ok as it introduced the types of waste that could be used for biochar preparation. What is MS and RH, you must extension them when they appear firstly. Merge the 3rd and 4th paragraphs together because they were focusing on the same thing of biochar modification. Here, we high suggested the author to cite https://doi.org/10.3390/polym15122741 . It is really detailed describing the modification approach of biochar material.
(4) In the second section of the waste biomass feedstock. actually, the feedstocks for biochar is biomass. The authors have further classified biomass into three really detailed types. It is good, but the author needs to provide their own opinions on biochar feedstocks, and the different performance of biochar from various sources.
(5) The title of the third section should be revised as biochar modification technology NOT modified biochar as this section systematically reviewed the methods for biochar post modification. Also, the author needed to provide some conclusive sentences to state their opinions.
(6) Line 350. It should be 4. Environmental applications of biochar. The manuscript must be carefully gone through before submission.
(7) From the topic map, I cannot see any emphasis on “removal of pollutants, chemicals and pharmacy components”. Therefore, the topic map need to be revised.
(8) The conclusion part is not concise, I hope the author could briefly summarized their work and give some constructive perspectives here. In addition, catalytic material with high adsorption properties may caused confused in experiment as it is difficult to distinguish the adsorption fraction and the degradation fraction. But it is not an obstacle, because it could concentrated the pollutants around the material and promoted a high performance degradation.
Minor comments
(1) The English quality need to be improved.
(2) Figures and tables using inconsistent styles including the font types and sizes. Need to consider topic map, Figure 1, Table 2 and 3, et al.
(3) Modification methods is better than modified methods in text and figures.
(4) The whole manuscript must be carefully gone through before submission as there are some mistakes.
I hope the author could kindly address these comments in the revised manuscript to meet the requirements of publication. My suggestion is this manuscript could be considered to be accepted if the issues were solved.
English need to be improved. Better to ask some prefessional edition service or seek the help from native speaker.
Author Response
Thank you for your useful suggestions and reviewers' and the editors comments on our manuscript. We have modified the manuscript accordingly, and detailed corrections are in red in the revised manuscript, the answers to reviewers' and editor s' questions are listed below point by point:

Reviewer 4 Report (New Reviewer)
Reviewer Comments
1. Highlights are too generic, specific highlights need to be mentioned.
2. The present article focused on Feedstocks, modification and application. It is recommended to include a separate section for the Biochar production techniques (Pyrolysis, Gasification, Hydrothermal Carbonization, etc., e.g.: https://doi.org/10.1007/s13204-021-01924-2)
3. The introduction needs to be strengthened by addressing the need for the research and importance of biochar globally (https://doi.org/10.1002/slct.201902127)
4. In section 2.1, still more recent research works are recommended to include. Some of the recent research work are listed below. In addition to the mentioned research, authors are free to include other relevant research
https://doi.org/10.30955/gnj.004491
https://doi.org/10.30955/gnj.004492
https://doi.org/10.30955/gnj.004496
https://doi.org/10.30955/gnj.004502
https://doi.org/10.30955/gnj.004497
https://doi.org/10.30955/gnj.004061
https://doi.org/10.1016/j.jics.2021.100107
5. In section 2.4. For nontraditional materials, it is recommended to include the different feedstocks that are used for biochar production. Some of the reference authors can be used from the list below.
10.5004/dwt.2022.28667
https://doi.org/10.1155/2022/7323588
https://doi.org/10.1016/j.chemosphere.2021.132368
https://doi.org/10.1155/2021/6397137
https://doi.org/10.1155/2021/1535823
https://doi.org/10.5004/dwt.2020.25339
https://doi.org/10.1002/wer.1092
https://doi.org/10.1016/j.jece.2019.103297
https://doi.org/10.1080/15226514.2019.1612845
https://doi.org/10.1080/15567036.2021.1943070
https://doi.org/10.1007/s13399-021-01483-0
https://doi.org/10.1007/s13399-020-01268-x
6. Table 3 needs to be elaborated more, it is recommended to include a column representing the method used for biochar production and its pyrolysis temperature. It is also recommended to include some more feedstocks and removal efficiency of different pollutants from the above mentioned references.
7. In conclusion, it is recommended to include only important findings and futuristic applications. The present conclusion is too long and needs to be concise.
Author Response
Thank you for your useful suggestions and reviewers' and the editors comments on our manuscript. We have modified the manuscript accordingly, and detailed corrections are in red in the revised manuscript, the answers to reviewers' and editor s' questions are listed below point by point:

Round 2
Reviewer 1 Report (New Reviewer)
Suitable corrections were made. I accept this manunuscript in present form.
Reviewer 2 Report (New Reviewer)
The paper has been well modified and is now suggested for publication. Good luck,
Reviewer 4 Report (New Reviewer)
Accept
This manuscript is a resubmission of an earlier submission. The following is a list of the peer review reports and author responses from that submission.
Round 1
Reviewer 1 Report
Describes the application of biochar prepared from waste in the environmental field, which is rich in research and at the same time the research is valuable and acceptable. However, from the manuscript, there are still some problems with this work's , which need to be revised by the authors.
The specific modifications are as follows:
1.Abstract: It's too brief. The content and innovation of this work are not summarized.
2.Abstract: It's section needs to be specifically descriptive and summarise everything.
3. Introduction: Language needs to be improved, and this is a problem throughout the manuscript.
4.In the section on biochar applications there is a lot of content, which is not conducive to the reader's quick grasp of the information, and charts and graphs could be added appropriately.
5.Table 1 provides information but is not explicitly mentioned in the text. Consider making changes.
6.In the Carbon sequestration section you can add the latest research literature, such as Wu's findings.
7.This article is poorly organised and grammatically confusing and needs a professional to touch up the grammar.
8.Conclusion: this conclusion should add something. The conclusion of a review article should not only summarise the main points of this work, but also present additional points of view of its own, such as shortcomings of the current research and potential solutions.
1. The whole manuscript should pay attention to the unity of tense.
2. Try to use as many different words as possible to replace the same meaning.
Author Response
-Reviewer 1
Comments and Suggestions for Authors
Describes the application of biochar prepared from waste in the environmental field, which is rich in research and at the same time the research is valuable and acceptable. However, from the manuscript, there are still some problems with this work's , which need to be revised by the authors.
The specific modifications are as follows:
1.Abstract: It's too brief. The content and innovation of this work are not summarized.
Answer: L25-28
The feedstock for biochar includes four categories: agricultural and forestry waste, industrial byproducts, municipal solid waste and other non-traditional materials.
2.Abstract: It's section needs to be specifically descriptive and summarise everything.
Answer: L31-34
In addition to being widely used as an adsorbent, catalyst and activator, biomass charcoal also has good application prospects as a soil remediation agent, amendment agent and supercapacitor, and in soil carbon sequestration.
- Introduction: Language needs to be improved, and this is a problem throughout the manuscript.
Answer: Research on the preparation of biochar from waste and its application in the environmental field has undergone English language editing by MDPI. The text has been checked for correct use of grammar and common technical terms, and edited to a level suitable for reporting research in a scholarly journal. MDPI uses experienced, native English speaking editors. Full details of the editing service can be found at https://www.mdpi.com/authors/english.
English Editing ID:english-69649
4.In the section on biochar applications there is a lot of content, which is not conducive to the reader's quick grasp of the information, and charts and graphs could be added appropriately.
Answer: Add the table as follows
Table 4. Removing antibiotics from the environment using biochar.
|
Absorbent |
Biochar |
Maximum adsorption capacity |
References |
|
Fluoroquinolone antibiotics |
Sludge biochar modified with alkali and bimetallic salts |
55.7mg/g |
[91] |
|
Sulphamethoxazole |
Algae-derived biochar |
|
[92] |
|
Sulphamethoxazole |
Boric acid-activated biochar |
|
[93] |
|
Sulphamethoxazole |
FeCl3-activated bermudagrass-derived biochar |
|
[94] |
|
Tetracyclines |
Hydrothermal mesoporous biochar |
|
[95] |
5.Table 1 provides information but is not explicitly mentioned in the text. Consider making changes.
Answer: L272-276
In summary, the raw materials of biomass charcoal are mainly divided into five categories, but even for materials from the same source, the preparation process is not the same; the actual application of the material needs to be based on the material’s specific properties to control the vegetation time and temperature.
6.In the Carbon sequestration section you can add the latest research literature, such as Wu's findings.
Answer: L799-802
Other studies also used biochar to form an organic carrier to prepare a good CO2 adsorbent with an adsorption capacity of 4.23 mmol CO2-g-1 and its performance remained stable after 30 cycles of adsorption experiments [136]..
7.This article is poorly organised and grammatically confusing and needs a professional to touch up the grammar.
Answer: Research on the preparation of biochar from waste and its application in the environmental field has undergone English language editing by MDPI. The text has been checked for correct use of grammar and common technical terms, and edited to a level suitable for reporting research in a scholarly journal. MDPI uses experienced, native English speaking editors. Full details of the editing service can be found at https://www.mdpi.com/authors/english.
English Editing ID:english-69649
8.Conclusion: this conclusion should add something. The conclusion of a review article should not only summarise the main points of this work, but also present additional points of view of its own, such as shortcomings of the current research and potential solutions.
Answer: L856-859; L874-877
In practical applications, selecting the appropriate raw materials, preparation process and modification means according to the specific situation and the state being explore is required, and this is an area with need of future research.
When biomass charcoal is used as a catalyst, since it also has the function of adsorption, it is important to determine whether it is catalytic or adsorptive in a specific application.
Comments on the Quality of English Language
- The whole manuscript should pay attention to the unity of tense.
Answer: Research on the preparation of biochar from waste and its application in the environmental field has undergone English language editing by MDPI. The text has been checked for correct use of grammar and common technical terms, and edited to a level suitable for reporting research in a scholarly journal. MDPI uses experienced, native English speaking editors. Full details of the editing service can be found at https://www.mdpi.com/authors/english.
English Editing ID:english-69649
2.Try to use as many different words as possible to replace the same meaning.
Answer: Research on the preparation of biochar from waste and its application in the environmental field has undergone English language editing by MDPI. The text has been checked for correct use of grammar and common technical terms, and edited to a level suitable for reporting research in a scholarly journal. MDPI uses experienced, native English speaking editors. Full details of the editing service can be found at https://www.mdpi.com/authors/english.
English Editing ID:english-69649
Reviewer 2 Report
First of all, the authors should elaborate Introduction section and thoroughly provide a state-of-art of the current problem related to the environmental application of BC. There are a lot of previously published nice reviews on a similar topic, and it a quietly important to demonstrate the relevance and significance of the present review paper. Please use more relevant and authoritative papers for state-of-art discussion (10.1007/s42773-023-00207-z, 10.1016/j.aoas.2019.12.006, 10.1016/j.biortech.2017.08.122 et al)
Appropriate references should support all figures and schemes from the other studies.
Data presented in the Table #3 should be extended (please add the amount of utilized sorbent, pH, and the method of BC preparation/modification)
Please add a similar comparative table for the catalytic application of the BC.
I suggest adding an extra section related to the nanoparticles-modified BC and its environmental applications to this review. This is an extremely new way of BC applications, and it will be interesting for wide spectra of readers.
Author Response
-Reviewer 2
Comments and Suggestions for Authors
First of all, the authors should elaborate Introduction section and thoroughly provide a state-of-art of the current problem related to the environmental application of BC. There are a lot of previously published nice reviews on a similar topic, and it a quietly important to demonstrate the relevance and significance of the present review paper. Please use more relevant and authoritative papers for state-of-art discussion (10.1007/s42773-023-00207-z, 10.1016/j.aoas.2019.12.006, 10.1016/j.biortech.2017.08.122 et al)
Answer: L100-108
Overall, the type of feedstock and preparation conditions are determining factors for the productivity of biochar[21]. The main application of biochar is for pollutant removal, followed by climate change mitigation, soil fertility improvement, waste management and atmospheric carbon sequestration into the soil[22]. Then, when biochar is used to remove pollutants, this mainly relies on the functional groups of biochar such as carboxyl and hydroxyl groups, which have different degradation mechanisms for different pollutants, some due to ion exchange and others due to surface complexation[23].
Appropriate references should support all figures and schemes from the other studies.
Answer: L272-276
In summary, the raw materials of biomass charcoal are mainly divided into five categories, but even for materials from the same source, the preparation process is not the same; the actual application of the material needs to be based on the material’s specific properties to control the vegetation time and temperature.
Data presented in the Table #3 should be extended (please add the amount of utilized sorbent, pH, and the method of BC preparation/modification)
Answer: Readjust the graphic abstract as shown below
Table 3 Removal of heavy metals from the environment with biochar
|
Heavy metals |
Initial concentration |
Material |
Modification methods
|
Maximum adsorption capacity |
Removal efficiency |
References |
|
Ni(II) |
20 mg/L |
Peanut shell |
Load strains |
|
81.17% |
[70] |
|
Cu(II) |
20 mg/L |
Peanut shell |
Load strains |
|
45.84% |
[70] |
|
Cr(VI) |
20 mg/L |
Peanut shell |
Load strains |
|
38.21% |
[70] |
|
Sb(III) |
40 mg/L |
Pteridium aquilinum |
Loading chitosan |
168 mg/g |
88% |
[72] |
|
Sb(V) |
24.36mg/L |
Grain |
Phosphogypsum modification |
8123 mg/kg |
|
[73] |
|
Cd(II) |
50mg/L |
Eastern Maple Leaf |
Oxidiser modification |
52.5 mg/g |
98.57% |
[74] |
|
Cd(II) |
50mg/L |
Apple branch |
Phosphate modification |
116 mg/g |
99.98% |
[75] |
|
T1(I) |
|
wheat straw |
|
382.38 mg/g |
90% |
[77] |
|
Pb(II) |
|
Corncob |
Sulphur modification |
421.8 mg/g |
|
[78] |
|
Pb(II) |
|
Wheat straw |
Magnetisation
|
817.64mg/g |
|
[79] |
|
Pb(II) |
600 mg/L |
Peanut shell |
KHCO3 activation and MgO nanoparticles incorporation |
1625.5 mg/g |
|
[80] |
|
Cd(II) |
100 mg/L |
Peanut shell |
KHCO4 activation and MgO nanoparticles incorporation |
480.8 mg/g |
|
[80] |
|
Cd(II) |
500 mg/L |
Rice husk biochar |
Introduction of specific functional groups |
195.5mg/g |
|
[81] |
|
Pb(II) |
500 mg/L |
Rice husk biochar |
Introduction of specific functional groups |
198.93mg/g |
|
[81] |
|
Pb(II) |
|
Coconut shell |
In situ formation of zinc oxide templates |
|
|
[82] |
|
Cd(II) |
|
Coconut shell |
In situ formation of zinc oxide templates |
|
|
[82] |
|
Pb(II) |
200mg/L |
Laminated wood |
Sulfonation
|
191.07 mg/g |
|
[83] |
|
Cd(II) |
100mg/L |
Laminated wood |
Sulfonation
|
85.76 mg/g |
|
[83] |
Please add a similar comparative table for the catalytic application of the BC.
Answer: L677-678
Meanwhile, Table 5 summarises the information on biochar when used as a photocatalyst in the environment.
Table 5. Biochar in the environment as a photocatalyst.
|
Degradable |
Material |
Photocatalyst |
Degradation rate |
References |
|
Methylene blue |
Bamboo |
ZnO/biochar |
95.19% |
[104] |
|
Organic phosphorus pesticides |
Rice straw |
biochar/α-Fe2O3/MgO
|
82.3% |
[115] |
|
Carbamazepine |
Corncob |
Fe3O4/BiOBr/CQDs |
99.52% |
[117] |
|
Carbamazepine |
Reed straw |
Fe3O4/BiOBr/BC |
51.50% |
[118] |
|
Sulfamethoxazole |
Reed straw |
Zn-TiO2/pBC
|
81.21% |
[119] |
I suggest adding an extra section related to the nanoparticles-modified BC and its environmental applications to this review. This is an extremely new way of BC applications, and it will be interesting for wide spectra of readers.
Answer: L715-723
In addition to the above, common photocatalysts combining biomass charcoal and Tio2, there are also nanoparticles composited with them to make new hybrid nanomaterials. This is a combination of the advantages of both; the composite will have the ability to exhibit good surface properties, high porosity and stable chemical properties and will have higher photocatalytic capacity[120]. However, such composites may indirectly release toxic substances. Therefore, biodegradable and environmentally non-toxic materials should be selected for specific applications in the future[121].
Reviewer 3 Report
The submitted manuscript represents a review on the recent findings related with the biochar research including preparation, modification and applications. The specific subject is rather interesting to be examined by a review study in order to summarize significant achievements in the field and give an overview of the advantages and shortcoming of the technology based on this material. Unfortunately, the current work fails to serve in this direction since it gives a very brief description of the sections covering only a minimum number of representative research examples. Apart from this, the manuscript is not well-written considering the grammar and syntax in many points and this makes understanding by the reader very difficult. To my opinion, the manuscript should not be considered for publication. Significant improvement of the linguistic part and careful analysis of current state of the art in the field providing something more than existing review studies is required.
The manuscript is not well-written considering the grammar and syntax in many points and this makes understanding by the reader very difficult.
Author Response
-Reviewer 3
Comments and Suggestions for Authors
The submitted manuscript represents a review on the recent findings related with the biochar research including preparation, modification and applications. The specific subject is rather interesting to be examined by a review study in order to summarize significant achievements in the field and give an overview of the advantages and shortcoming of the technology based on this material. Unfortunately, the current work fails to serve in this direction since it gives a very brief description of the sections covering only a minimum number of representative research examples. Apart from this, the manuscript is not well-written considering the grammar and syntax in many points and this makes understanding by the reader very difficult. To my opinion, the manuscript should not be considered for publication. Significant improvement of the linguistic part and careful analysis of current state of the art in the field providing something more than existing review studies is required.
Answer: L166-170
Wheat, the world's third largest cereal, is also a great source of biochar feedstock due to its malt roots. It can be used to make supercapacitor electrodes, which have increased supercapacitance values because of the material's good specific surface area[33,34].
Comments on the Quality of English Language
The manuscript is not well-written considering the grammar and syntax in many points and this makes understanding by the reader very difficult.
Answer: Research on the preparation of biochar from waste and its application in the environmental field has undergone English language editing by MDPI. The text has been checked for correct use of grammar and common technical terms, and edited to a level suitable for reporting research in a scholarly journal. MDPI uses experienced, native English speaking editors. Full details of the editing service can be found at https://www.mdpi.com/authors/english.
English Editing ID:english-69649
Reviewer 4 Report
WATER 2538942
Research on the preparation of biochar from waste and its application in the environmental field
Wanyue Wang , Jiacheng Huang, Tao Wu , Xin Ren, Xuesong Zhao
This review is about different BC produced from different raw waste materials and methods to treat the BC. Then the review summarizes the environmental applications of BC
The review is not well written. It is a simple quote of some articles without critical look. A lot of references, some highly cited, are missing in the parts where BC from different sources was discussed.
At the end of chapter 2, I was expecting a paragraph where the different BC were compared and discussed. Even at the end of chapter 3 the paragraph about the comparison of different methods for post treatment is not useful. Generally the authors refer only to one study in each case and this is not in favor of the review.
I am sorry but I cannot propose acceptance or revision of this review. I suggest the authors to cover only a part of the listed issues and focus on it, trying to be more critical and compare the results from different articles.
Author Response
-Reviewer 4
Comments and Suggestions for Authors
WATER 2538942
Research on the preparation of biochar from waste and its application in the environmental field
Wanyue Wang , Jiacheng Huang, Tao Wu , Xin Ren, Xuesong Zhao
This review is about different BC produced from different raw waste materials and methods to treat the BC. Then the review summarizes the environmental applications of BC
The review is not well written. It is a simple quote of some articles without critical look. A lot of references, some highly cited, are missing in the parts where BC from different sources was discussed.
Answer: L166-170
Wheat, the world's third largest cereal, is also a great source of biochar feedstock due to its malt roots. It can be used to make supercapacitor electrodes, which have increased supercapacitance values because of the material's good specific surface area[33,34].
At the end of chapter 2, I was expecting a paragraph where the different BC were compared and discussed. Even at the end of chapter 3 the paragraph about the comparison of different methods for post treatment is not useful. Generally the authors refer only to one study in each case and this is not in favor of the review.
Answer: L122-130
Of these four different sources of materials, agricultural and forestry wastes are the most readily available and abundant, so the advantages of biomass charcoal produced from them are ease of preparation, large quantities, and low production costs at low prices. Industrial byproducts and other byproducts generated in the production process can also be used to generate additional income for the factory, which is a greater realisation of the reuse of waste resources. The variety of municipal solid waste and other non-traditional materials is not conducive to uniform treatment.
I am sorry but I cannot propose acceptance or revision of this review. I suggest the authors to cover only a part of the listed issues and focus on it, trying to be more critical and compare the results from different articles.
Editorial Summary:
Academic Editor Notes
Thank you for submitting your manuscript to our special issue. We appreciate the time and effort you have invested in this work.
The manuscript has been thoroughly reviewed by four referees, and their comments reflect diverse opinions. While some aspects of the manuscript were praised, there were significant concerns raised about its structure, language, and content.
Should the editorial decision lean towards a major revision or reject or resubmission, I suggest focusing on the following areas for improvement:
Language and Grammar: Consider employing a professional language service to ensure clarity and grammatical accuracy throughout the manuscript.
Answer: For grammar and other problems in the language, there is a treatment: find a professional touch-up artist to do the touch-up treatment.
Organization and Content Flow: Enhance the structure of the paper, focusing on clear transitions and logical connections between sections.
Inclusion of Supporting Data: Consider adding more tables, charts, or graphs to aid the reader's understanding, particularly in the section on biochar applications.
Answer: L677-678
Meanwhile, Table 5 summarises the information on biochar when used as a photocatalyst in the environment.
Table 5. Biochar in the environment as a photocatalyst.
|
Degradable |
Material |
Photocatalyst |
Degradation rate |
References |
|
Methylene blue |
Bamboo |
ZnO/biochar |
95.19% |
[104] |
|
Organic phosphorus pesticides |
Rice straw |
biochar/α-Fe2O3/MgO
|
82.3% |
[115] |
|
Carbamazepine |
Corncob |
Fe3O4/BiOBr/CQDs |
99.52% |
[117] |
|
Carbamazepine |
Reed straw |
Fe3O4/BiOBr/BC |
51.50% |
[118] |
|
Sulfamethoxazole |
Reed straw |
Zn-TiO2/pBC
|
81.21% |
[119] |
Comprehensive Analysis: Ensure a more critical and comparative analysis of the literature, especially in areas where biochar from different sources is discussed.
Unique Contribution and Emphasis on Water Environmental Applications: Emphasize what sets this review apart from existing studies, and consider adding a section on innovative applications, such as nanoparticles-modified BC. Strengthen the focus on the application of biochar in the water environment, aligning with the theme of our special issue.
Answer: L715-723
In addition to the above, common photocatalysts combining biomass charcoal and Tio2, there are also nanoparticles composited with them to make new hybrid nanomaterials. This is a combination of the advantages of both; the composite will have the ability to exhibit good surface properties, high porosity and stable chemical properties and will have higher photocatalytic capacity[120]. However, such composites may indirectly release toxic substances. Therefore, biodegradable and environmentally non-toxic materials should be selected for specific applications in the future[121].
Please understand that these suggestions are contingent on the final editorial decision, and you will be informed accordingly by the editor-in-charge.
Thank you once again for your valuable contribution, and please do not hesitate to reach out if you have any questions.
The manuscript has been resubmitted to your journal. We look forward to your positive response.
Round 2
Reviewer 2 Report
The authors replied to all my queries, thus in the present form, this review can be recommended for publication.
Reviewer 3 Report
Not much improvement concerning the previous comments. I will stay in my initial opinion.
Already checked by MDPI service
Reviewer 4 Report
In my first review I asked the authors to b more critical and put some effort in the comparison of different raw biomass and the corresponding biochars. Without this I cannot find the novelty and the value of the contribution. Even if the scope is only to record different efforts the current version of the manuscript fails to support it. The revision of the authors is not satisfuctory and I am sorry but I have to reject it